# Hyperbranched Polycaprolactone through RAFT Polymerization of 2-Methylene-1,3-dioxepane

**DOI:** 10.3390/polym11020318

**Published:** 2019-02-13

**Authors:** Ping Xu, Xiaofei Huang, Xiangqiang Pan, Na Li, Jian Zhu, Xiulin Zhu

**Affiliations:** 1State and Local Joint Engineering Laboratory for Novel Functional Polymeric Materials, Jiangsu Key Laboratory of Advanced Functional Polymer Design and Application, Department of Polymer Science and Engineering, College of Chemistry, Chemical Engineering and Materials Science, Soochow University, Suzhou 215123, China; 20164209052@stu.suda.edu.cn (P.X.); huangxiaofei314@163.com (X.H.); panxq@suda.edu.cn (X.P.); xlzhu@suda.edu.cn (X.Z.); 2Jiangsu Litian Technology Co. Ltd., Rudong County, Jiangsu 226407, China; 3Global Institute of Software Technology, No 5. Qingshan Road, Suzhou National Hi-Tech District, Suzhou 215163, China

**Keywords:** polycaprolactone, hyperbranch, living radical polymerization

## Abstract

Hyperbranched polycaprolactone with controlled structure was synthesized by reversible addition-fragmentation chain transfer radical ring-opening polymerization along with self-condensed vinyl polymerization (SCVP) of 2-methylene-1,3-dioxepane (MDO). Vinyl 2-[(ethoxycarbonothioyl) sulfanyl] propanoate (ECTVP) was used as polymerizable chain transfer agent. Living polymerization behavior was proved via pseudo linear kinetics, the molecular weight of polymers increasing with conversion and successful chain extension. The structure of polymers was characterized by ^1^H NMR spectroscopy, tripe detection gel permeation chromatography, and differential scanning calorimetry. The polymer composition was shown to be able to tune to vary the amount of ester repeat units in the polymer backbone, and hence determine the degree of branching. As expected, the degree of crystallinity was lower and the rate of degradation was faster in cases of increasing the number of branches.

## 1. Introduction

Aliphatic polyesters have received much attention due to their good biocompatibility and biodegradability. Typical aliphatic polyesters, such as polyglycolide (PGA), polylactide (PLA), and polycaprolactone (PCL) [1], have been extensively researched in the past decades. Among them, PCL has shown a good balance of biocompatibility, thermoplasticity, biodegradability, crystallization, and non-woven, etc. Materials based on PCL are widely used in drug release coatings, shape memory materials, and various material modifiers [2,3,4,5]. PCL was firstly synthesized by Carothers’ group [6] in the 1930s through condensation polymerization. However, due to the limitation of condensation polymerization, most PCL used nowadays is generally prepared by ring-opening polymerization of caprolactone (CL), through cationic polymerization, anionic polymerization, coordination-insertion polymerization, and enzyme-catalyzed polymerization. Alternatively, Bailey’s group [7] reported, for the first time, that PCL-like polymers were prepared by free radical ring-opening polymerization of monomeric 2-methylene-1,3-dioxepane (MDO) (Scheme 1) in 1982. They successfully achieved the homopolymerization of MDO and its copolymerization with vinyl monomers, such as styrene and methyl methacrylate (MMA), using azodiisobutyronitrile (AIBN) and di-*t*-butyl peroxide (DTBP) as initiators.

2-Methylene-1,3-dioxepane (MDO), which is also called cyclic ketene acetals (CKAs), can undergo radical addition on their carbon–carbon double bond, which then subsequently leads to propagation, either by ring opening or ring retaining or by a combination of both, which depends especially on their structure (Scheme1) [8]. This monomer provides a novel and alternative synthetic route for the synthesis of PCL, and it imparts degradability to conventional vinyl monomers by introducing ester linkages into the backbone [9]. Thereby, the structure of the polyester material can be greatly expanded, and the polyester material can be applied to a wider range of fields. Thus, the free radical ring-opening polymerization of this monomer has attracted considerable interest of quite a few researchers [10].

In the last two decades, researchers found out that living controlled radical polymerization can also be introduced into rROP polymerization, including nitrogen–oxygen stable radical ring-opening polymerization (NMP) [11,12,13], reversible addition cleavage chain transfer ring-opening polymerization (RAFT) [8,14,15,16], atom transfer radical ring opening polymerization (ATRP) [17,18,19,20], and photo-induced cobalt-mediated radical polymerization (CMRP) [21]. This new type of living radical ring-opening polymerization can effectively control the molecular weight and molecular weight distribution of the polymer, thereby imparting various properties to the polymer. There have been many successful cases regarding the free (living) radical copolymerization of MDO with different vinyl monomers, including a large number of hydrophobic monomers, such as styrene (St) [22,23], methyl methacrylate (MMA) [24,25], methyl acrylate (MA) [26], glycidyl methacrylate (GMA) [27,28], 2-vinyl-4,4-dimethyl azlactone (VDMA) [29], vinyl acetate (VAc) [30,31], and hydrophilic monomers, such as polyethylene glycol dimethyl ether methyl methacrylate (PEGMA) [32], *N*,*N*-dimethyl methacrylate (DMAEMA) [33], *N*-vinyl pyrrolidone (NVP) [34,35], *N*-isopropyl acrylamide (NIPA) [36], etc. Besides, in the past two years, MDO has also been used to copolymerize with some protein resistible monomers for use in marine antibiofouling by Zhang’s group [37,38,39].

It is well known that the performance of polymer is closely related to its structure, while the topology of polymer is an important part of its chain structure. According to the connection modes, the polymer topology can be divided into grafting, block, random and alternating copolymer, star, annular, dendritic, hyperbranched polymer and polymer brush, etc. Polymers with nonlinear structures have a lower melting viscosity than linear polymers, allowing them to process at lower temperatures. Recently, dendritic branched macromolecules have become a hot topic in many fields [40,41,42]. Especially, hyperbranched polymer was thought to have wide industry prospects due to its simple synthesis method [43]. Furthermore, hyperbranched polymer has more advantages compared with the two-block or three-block polymers, such as the ability to form a small single molecule micelle, lower viscosity, reduce the molecular chain tangles, etc. [44].

There are many preparation methods of hyperbranched polymer. The self-condensing vinyl copolymerization (SCVP), using a species capable of both propagation and initiation, can efficiently prepare highly branched polymer [45]. To the best of our knowledge, no attempts have been made to synthesize hyperbranched polycaprolactone using MDO through one-pot RAFT polymerization. In this study, we synthesized a RAFT reagent with a double bond at one end. The hyperbranched polycaprolactone was synthesized by one-pot polymerization of MDO, and the relationship between the degree of branching, crystallinity, and degradation rate was studied.

## 2. Materials and Methods

### 2.1. Materials

Vinyl acetate (VAc) (>99%, Aldrich, Shanghai, China) was purified by passing through a neutral alumina column and then stored at −4 °C. 2,2-Azobisisobutyronitrile (AIBN) (AR, Shanghai Chemical Reagents Co. Ltd., Shanghai, China) was purified by recrystallization from ethanol. Bromoacetaldehyde dimethyl acetal (Adamas, Shanghai, China), Dowex 50 (Alfa Aesar, Shanghai, China), Aliquat 336 (J&K, Shanghai, China), potassium tert-butoxide (Energy Chemical, Suzhou, China), 2-bromopropionic acid (98%, Alfa Aesar), and palladium(II) acetate (Pd 46%~48%, Macklin, Shanghai, China) was used as received. All other chemicals were obtained from Shanghai Chemical Reagents Co. Ltd., Shanghai, China, and used as received. 2-Methylene-1,3-dioxepane (MDO) and RAFT-agent 2-Ethoxythiocarbonylsulfanyl-propionic acid ethyl ester (EXEP) and [(ethoxycarbonothioyl) sulfanyl] propanoate (ECTVP) was synthesized according to previously published references [7,46,47] (see Scheme 2). The NMR spectra are shown in Appendix A.

### 2.2. Synthesis of Linear PCL

A typical polymerization procedure, using AIBN as an initiator and EXEP as a chain transfer agent, was carried out at 60 °C with a molar ratio of [MDO]_0_:[EXEP]_0_:[AIBN]_0_ = 50:1:0.2. A mixture of MDO (5 mmol, 572 mg), EXEP (0.1 mmol, 22.2 mg), and AIBN (0.02 mmol, 3.2 mg) was placed in a dried ampule with a stir bar. The content was degassed by three freeze–evacuate–thaw cycles. The ampoule was flame-sealed and placed into heating bath thermoset at 60 °C. The ampule was taken out and opened after 24 h. The crude product was dissolved in THF and precipitated into a large amount of cold methanol. The polymer was obtained by filtration and dried under vacuum at 30 °C until constant weight. The conversion was determined through gravimetric. The molecular weight and molecular weight distribution were determined by gel permeation chromatography (GPC).

### 2.3. Synthesis of Hyperbranched PCL

A typical polymerization procedure using AIBN as an initiator and ECTVP as a chain transfer agent was carried out at 60 °C with a molar ratio of [MDO]_0_:[ECTVP]_0_:[AIBN]_0_ = 100:1:0.2. A mixture of MDO (10 mmol, 1.14 g), ECTVP (0.1 mmol, 21.2 mg), and AIBN (0.02 mmol, 3.3 mg) was placed in a dried ampule with a stir bar. The content was degassed by three freeze–evacuate–thaw cycles. The ampoule was flame-sealed and placed into a heating bath thermoset at 60 °C. At different time intervals, the ampule was taken out and opened. The crude product was dissolved in THF and precipitated into a large amount of cold methanol. The polymer was obtained by filtration and dried under vacuum at 30 °C until constant weight. The conversion was determined through gravimetric. The molecular weight and molecular weight distribution were determined by GPC.

### 2.4. Chain Extension

MDO (5 mmol, 572 mg) or VAc (5 mmol, 434 mg), macro-CTA (*M*_n_ = 5100 g mol^−1^, *Ð* = 2.09) (0.05 mmol, 275 mg) and AIBN (0.01 mmol, 1.7 mg) were added into a dried ampule with a stir bar. The content was degassed by three freeze–evacuate–thaw cycles. The ampoule was flame-sealed and transferred into a stirring apparatus and polymerized at 60 °C for 24 h. Subsequently, the ampule was taken out and opened. The crude product was dissolved in THF, precipitated into a large amount of cold methanol and then stored at −18 °C. The polymer was obtained by filtration and dried under vacuum at 30 °C until constant weight.

### 2.5. Degradation

30 mg of the copolymers were dissolved in a small amount of methylene dichloride. A solution of KOH in methanol (0.025 M, 5 mL) was then added to the vial and stirred at 40 °C. After time interval, the samples were taking out from vial and evaporating under vacuum. The residual solid was dissolved in THF for GPC determination.

### 2.6. Characterization

^1^H NMR spectroscopy was performed on a Bruker 300 MHz nuclear magnetic resonance instrument. The number-average molecular weight (*M*_n,GPC_) and molecular weight distribution (*Đ*) of the polymers were determined with Waters 1515 gel permeation chromatography, equipped with triple detection, including Waters 2424 refractive index detector, Wyatt VicoStar viscosity detector, and Wyatt TRI STAR Mini DAWN HELEOS II eighteen angle light scattering detector, using HR 1, HR 2, and HR 4 (7.8 × 300 mm^2^, 5 μm beads size) columns with a measurable molecular weight ranged 5 × 10^2^ to 5 × 10^5^ g mol^−1^. THF was used as the eluent at a flow rate of 0.6 mL min^−1^ and 40 °C. GPC samples were injected using a TOSOH plus auto sampler and calibrated with PS standards purchased from TOSOH. Thermogravimetric analysis (TGA) was carried out on a 2960 SDT TA instruments with a heating rate of 10 °C min^−1^ from 30 to 800 °C under the nitrogen atmosphere. The DSC analyses for these polymers were performed on a TA Instrument DSC Q200from −80 to 80 °C under a nitrogen atmosphere, with a heating rate and a cooling rate of 20 °C min^−1^ for the first time, and a heating rate of 10 °C min^−1^ for the second time. The degree of crystallinity of the polymers was measured in the DSC-thermogram of the crystallized peak (exotherm peak) and the molten peak (endothermal peak) in the second heating cycle.

## 3. Results and Discussion

### 3.1. Polymerization Behavior of MDO in the Presence of ECTVP

Radical ring opening polymerization of MDO in the presence of ECTVP, using AIBN as the initiator, was investigated with different feeding ratios of monomer and chain transfer agent, at 60 °C. The conversion along with molecular weight of the obtained polymers and their distribution were summarized in Table 1 and Figure 1. The results showed that polymerization could be carried out with reasonable conversion under these conditions. The monomer conversion decreased from 99.3% to 48.2% after 96 h of polymerization, with the feeding ratio of MDO to ECTVP increasing from 10 to 200, which was mostly due to the decreasing of initiator concentration. It showed that no obvious relationship between the feeding ratio and molecular weight of the obtained polymer, which may due to multiple effects combined in such a polymerization, such as feeding ratio, conversion, and initiator concentration. However, the molecular weight distribution of the polymer became narrower in cases of higher MDO molar ratio.

It was known that a hyperbranched structure could be obtained by using ECTVP as the RAFT agent, due to it containing a polymerizable double bond. Thus, the broad molecular weight distribution of the polymer may be related to the hyperbranched structure. In order to verify the hyperbranched structure, the obtained polymers were further characterized using triple detection GPC. For comparison, PCL was also prepared through radical ring opening polymerization of MDO, using EXEP as the RAFT agent under the same conditions (Entry P5 in Table 1, NMR spectrum of P5 was showed in Appendix A). The branching factors (*g’*) are defined as the intrinsic viscosity ratio of the branched polymer to the linear polymer at the same molecular weight. Based on the unique structure of the branched polymer, the lower the branching factors, the higher the degree of branching. The branching factors (*g’*) of these polymers can be calculated by the following equation, *g’* = η(branched)/η(linear), on the basis of the triple detection GPC, which were summarized in Table 1. The *g’* value of PCL from EXEP was 0.998, lower than 1, which verified the ease of branching in ring opening polymerization of MDO [48]. The *g’* value of PCLs from ECTVP were much lower than 1, especially in cases of low MDO feed ratio, which implied the hyperbranched structure existed in these polymers. Additionally, the broad peak, due to the merge of multi peaks in GPC traces (Figure 1), further verified such hyperbranched structures. was normal occurred in hyperbranched polymers. Thus, hyperbranched PCL could be prepared by using ECTVP as the RAFT agent through SCVP route. The degree of branching could be changed by changing the feeding ratio of ECTVP in polymerization.

### 3.2. Structure Analysis and Thermal Property

The structure of the obtained polymer was characterized by ^1^H NMR spectroscopy (Figure 2). Characteristic resonance peaks derived from MDO could be found with chemical shift of 3.98, 2.27, 1.53, 1.29 ppm, labelled as h, i, j, b, and l in Figure 2. Furthermore, a residual of terminal vinyl group with a chemical shift of 7.18 ppm and a thiocarbonyl-containing moiety with a chemical shift of 4.43 ppm, derived from ECTVP, could be found in the spectrum at of polymer c’ and d’, which agreed with the mechanism of SCVP. The branched structure of the obtained PCL could be verified by the existence of protons labelled as “i”, determined from ^1^H NMR analysis. The degree of branching (DB_NMR_) can be calculated on the basis of the ^1^H NMR data according to Equation (1) [49], and DB_theo_ can be calculated according to Equation (2) [50].
DB_NMR_ = 2[*b*/(*x* + 1)][1 − *b*/(*x* + 1)](1)
DB_theo_ = [2(1 − e^−(*x*+1))^(*x* + e^−(*x*+1)^)]/(*x* + 1)^2^(2)where, *x* is the molar ratio of [MDO]_0_ to [ECTVP]_0_, b is the fraction of branched units and linear units, and *B* is the fraction of initiating centers, which are satisfying the equation *b* + *B* = 1 [51]. Taking sample P1 in Figure 2 as an example, x value was 10, so the DB_theo_ could be calculated via Equation (2) to be 0.165. On the basis of integral values of the proton signal of methylene groups in the MDO units signals (b) at 4.04–3.93 (*I*_i_) and the menthine proton (B) at 4.39–4.34 (*I*_d’_), b was 0.996 in Figure 2, which could be calculated by formula b = *I*_i_/2/(*I*_i_/2 + *I*_d’_). As the result, the DB_NMR_ of sample P1 was 0.164, which was close to theoretical value (0.165). Other DB values of polymers obtained with different feeding ratios are listed in Table 2. It showed that as x increased from 10 to 200, the DB_NMR_ decreased from 0.164 to 0.01. The DB_NMR_ was close to DB_theo_, and the decreasing trend of the two were consistent.

Thermal stability of obtained PCL was studied by TGA at heating rate of 10 °C /min, from 30 to 800 °C, under nitrogen flow of 10 ml/min. The results were summarized in Figure 3. It showed that all of the hyperbranched polymers were stable under 250 °C, which was similar to linear PCL. The samples were erased of any thermal history by running the DSC for the first time at a heating rate of 20 °C /min, and then cooled before the DSC test in the second temperature-rising procedure. The second cycle of heating run, at the rate of 10 °C min^−1^, was adopted. The obtained DSC curves are shown in Figure 4. It is interesting to note that there were two obvious melting peaks in the thermograms of P3 and P4, while no melting peaks in the thermogram of P1, and a crystallization exotherm and a melting peak occurred in the thermogram of P2 at the same time. It is well known that the branched structure of polymers will lead to a decrease of crystallinity. It showed that P1 was free of crystal due to the highly branched structure. The crystallinity of P2 was the lowest among all samples, leading to its incomplete crystallization at room temperature. Then, the folding of the PCL segment became easier and further crystallization occurred when the samples were heated. So, both endothermal and exothermic peaks were displayed in the heating process of P2. The double melting peaks of P3 and P4 were claimed to be caused by the recrystallization of polymers and the subsequent melting.

The degree of crystallinity (χ) can be calculated by the following equations on the basis of the DSC trace [52,53], where χ(blend) is the crystallinity degree of the blend material.
χ(blend) = (Δ*Hm* − Δ*Hc*)/Δ*H_ƒ_*^0^χ = χ(blend)/ω(PCL)(3)

In Equation (3), Δ*Hm* is the heat of fusion of the polymer, which can be integrated from the endothermal peak, and Δ*Hc* is the heat of crystallization, which can be integrated from the crystallization exotherm during the same heating scan in the DSC thermograms. If there are no crystallization exotherms during the same heating scan, Δ*Hc* = 0. Besides, ω(PCL) is the weight fraction of PCL in the blend and Δ*H_ƒ_*^0^ = 136 J/g is the heat of fusion of 100% crystalline PCL [54].

According to the above method, we calculated the DB and χ of these polymers. These results are listed in Table 2. As can be seen from the table, all copolymers had a single glass transition temperature (*T_g_*) about −60 °C, which was similar to the pure polycaprolactone. Increasing the proportion of ECTVP incorporated into the copolymers, the degree of branching (DB) of the resultant polymer increased while the degree of crystallinity (χ) decreased. Besides, the branching factors (*g’*) and intrinsic viscosities (η) determined by GPC also increased with the increasing of *x*. When the *x* value was 200, *g’* was close to 1 and DB was close to 0, indicating that the structure of this polymer tended to be linear. Accordingly, this polymer had high crystallinity and characteristic viscosity. All this further indicates that the branching properties of the resulting copolymer were directly related to the amount of branching agent. So, excellent copolymers with different performance characteristics can be prepared by changing molar ratios by this method.

### 3.3. Copolymerization Kinetics

To further investigate the polymerization behavior, the kinetics of the RAFT/MADIX ROP polymerization of MDO was studied in different molar ratios, and is shown in Figure 5. Linear increase in the semilogarithmic kinetic plot was observed for all values of molar ratio. The polymerization rate increased with the decrease of x values, which was a result of the increase of AIBN concentration. Figure 6 indicates the dependence of the number-average molecular weight (*M*_n_) and the molecular weight distribution (*Ð*) on the monomer conversion at different concentrations of ECTVP. The molecular weight increased linearly with the monomer conversion in all cases of polymerizations. Thus, the polymerization kinetic and relationships, between molecular weight and conversion, showed the characteristic of controlled polymerization.

However, the obtained polymers showed broad molecular weight distribution. Both of the values of *M*_n_ and *Ð* were increased with the conversion. This has been shown to be the result of polymers with different hyperbranched structures formed during the synthesis of hyperbranched polymers.

### 3.4. Chain Extension

In order to verify there was retention of the dithioester moieties on the end of the hyperbranched PCL, extension of PCL (*M*_n_ = 5100 g mol^−1^, *Ð* = 2.4) with different monomers (MDO and VAc) under the condition of [monomer]_0_:[PCL]_0_:[AIBN]_0_ = 100:1:0.2 at 60 °C was performed. As expected, an obvious peak shift of the GPC trace toward a higher molar mass was observed, indicating that most chains were still living (Figure 7). The g’ factors of PCL-*b*-PVAc and PCL-*b*-PCL were 0.357 and 0.375, respectively, which were both smaller than 1, indicating that the polymer after chain extension was still hyperbranched.

### 3.5. Degradation

The degradability of hyperbranched and linear PCL was investigated by the hydrolysis of the copolymer sample in a solution of potassium hydrolysis (KOH, 0.025M) in methanol at 40 °C. In order to investigate the extent of degradability, here the GPC analyses were recorded at different time points (Appendix A). After the hydrolysis reaction, the disappearance of the main molar mass distribution originally by GPC analyses was observed, which showed that these polymers were hydrolytically degraded, as expected. It is interesting to note that the molecular weight of those hyperbranched polymers had an increasing process in a short time, which may be due to the fact that some of the low molecular weight polymers were degraded first, resulting in an increase of molecular weight. All of those’ degradation rates were very fast. In about 30 min, the polymers were basically degraded. Furthermore, in order to investigate the relationship between the extent of degradability and the DB of polymers, the molecular weight was recorded at different time points (Figure 8). The slope ($K_{p}^{\mathrm{app}}$) obtained at the linear part was the degradation rate of the polymer. It can be seen that the degradation rate of the linear PCL was the slowest according to the value of $K_{p}^{\mathrm{app}}$. Particularly, it was faster in the case of the polymer with a large value of DB, which can be found through Figure 8. All of these suggest that the degradability of the copolymer could be tuned by changing the DB of polymers.

## 4. Conclusions

In summary, hyperbranched PCL was successfully synthesized by SCVP-RAFT/MADIX polymerization in combination with radical ring opening polymerization of MDO. The kinetics and chain extension experiments showed that the polymerization process presented living characters. The degree of branching for the obtained hyperbranched PCL could be tuned by changing the molar ratio of MDO with ECTVP. Increasing the degree of branching will reduce the intrinsic crystallinity and increase the degradation rate, which offered a useful way to tune the properties of PCL.

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
