# Peer review of "Hyperbranched Polycaprolactone through RAFT Polymerization of 2-Methylene-1,3-dioxepane"

_polymers, 2019, doi:10.3390/polym11020318_

Round 1

Reviewer 1 Report

This article from the Zhu group shows nicely how degradable branched poylmers can be made using radical ring opening with RAFT.

I think this work can be published in polymer after some corrections.

In general the description of the results is a little brief. Expanding the data analysis would improve the work significantly.

P2 seems to have a exothermic peak. This needs to be addressed

The authors claim a Tg of -60 is seen. Looking at the raw DSC data does not make this clear. The Tgs are at best very weak.

Further explanation of how kpapp was determined for the degradation should be presented

Author Response

This article from the Zhu group shows nicely how degradable branched polymers can be made using radical ring opening with RAFT.

I think this work can be published in polymer after some corrections.

In general, the description of the results is a little brief. Expanding the data analysis would improve the work significantly.

Response: Thanks very much for your professional comments for improving the quality of our manuscript. The responses to your valuable comments have been made based on the results of supplemental experiments and a careful literature survey.

P2 seems to have a exothermic peak. This needs to be addressed.

Response: Thanks for the comments. The results showed that degree of crystallinity of sample P2 was lower than other samples, leading to its incomplete crystallization at room temperature. Then the folding of PCL segment became easier and further crystallization occurred when the samples were heated. So P2 displayed both endothermal and exothermic peaks in the heating process (J. Polym. Sci.: Part A: Polym. Chem., 2008, 46, 6486-6508). This part has been added to the article.

The authors claim a Tg of -60 is seen. Looking at the raw DSC data does not make this clear. The Tgs are at best very weak.

Reponses: Thanks for the comments. We are agreed that the Tg is showed weak signal in DSC data.

Further explanation of how kpapp was determined for the degradation should be presented

Response: Thanks for the comments. In order to investigate the relationship between the degradability and DB of polymers, the molecular weight was recorded at different time points (Fig. 8). The slope (Kappp) was obtained in the linear part. Such statement has been added into the revised manuscript.

Reviewer 2 Report

In this manuscript, XU et al. reported to use one-pot RAFT/MADIX polymerization of 2-Methylene-1, 3-dioxepane to prepare hyperbranched polycaprolactone. The kinetics of polymerization showed a controlled polymerization, which was also verified by chain extension. The polymers were characterized by TGA, DSC, GPC etc. Overall, the experiment was well designed with proper discussions. I recommend accepting in Polymers with minor revisions.

1. Table 1: Why higher ratio of [MDO]0: [ECTVP]0 resulted in a higher η(branched)/ η(linear)?

2.  Scheme 1 shows the radical polymerization of MDO has either by ring opening or ring retaining or by a combination of both products (Line 45). However, the authors considered all of them were ring-opening product (Figure 2). How to confirm this claim? If so, why there is no ring retaining product?

3. Line 220-221: “The double melting peaks of P3 and P4 were claimed to be caused by the recrystallization of polymers and subsequent melting.” What does “recrystallization of polymers and subsequent melting” mean? The two peaks are from melting. In addition, the authors should use an arrow to point the exo direction in DSC (Fig 4).

4. The polymer structure in Fig 2 is not correct. Please revise it.

Author Response

In this manuscript, XU et al. reported to use one-pot RAFT/MADIX polymerization of 2-Methylene-1, 3-dioxepane to prepare hyperbranched polycaprolactone. The kinetics of polymerization showed a controlled polymerization, which was also verified by chain extension. The polymers were characterized by TGA, DSC, GPC etc. Overall, the experiment was well designed with proper discussions. I recommend accepting in Polymers with minor revisions.

Q1: Table 1: Why higher ratio of [MDO]0: [ECTVP]0 resulted in a higher η(branched)/ η(linear)?

A1: Very thanks for your valuable question. Here, the value of η(branched)/ η(linear) means the branching factors (g’). Based on the unique structural of the branched polymer, the lower the branching factors, the higher the degree of branching. In this article, we have discussed the relationship between the ratio of [MDO]0: [ECTVP]0 (x) and the degree of branching (DB). As the x increase, the ECTVP contained in this reaction system decreases, so the DB decreases and the corresponding g’ increases.

Q2:  Scheme 1 shows the radical polymerization of MDO has either by ring opening or ring retaining or by a combination of both products (Line 45). However, the authors considered all of them were ring-opening product (Figure 2). How to confirm this claim? If so, why there is no ring retaining product?

A2: Fully agree and thanks very much for your profession suggestions. It is well known that there are three possible propagation routes during the radical polymerization of MDO. In this article, the obtained polymers were fully ring opened, which can be verified by the NMR spectrum of the polymer. If there is a ring retained structure, then the corresponding peaks 3.65-3.8 ppm (-CH2CH2O-) and 110 ppm (-CH2OCOCH2-) should be found in its 1H NMR and 13C NMR spectrum respectively. However, there were no peaks in Fig. 2 and Fig. 1A of our polymers, indicating that there was no acteal.

Fig. 1A 13C NMR spectrum of hyperbranched poly(ECTVP-co-MDO) in CDCl3 corresponding to entry P1 in Table 1 (Mn,GPC = 5100 g mol-1, Ð = 2.09, g’ = 0.548).

Generally speaking, the seven-membered monomer MDO has a strong steric effect on the non-ring-opened radicals, while providing greater stability to the ring-opened free radicals, which together promote the final fully open-loop polymer. (Polym. Chem., 2012, 3, 1260-1266.; Macromolecules, 1997, 30, 3104-3106.; J. Polym. Sci., Part A: Polym.Chem., 1982, 20,3021-3030)

Q3. Line 220-221: “The double melting peaks of P3 and P4 were claimed to be caused by the recrystallization of polymers and subsequent melting.” What does “recrystallization of polymers and subsequent melting” mean? The two peaks are from melting. In addition, the authors should use an arrow to point the exo direction in DSC (Fig 4).

A3: Thanks very much for your professional and positive comments. This senstence in quotes comes from Yuanqing Jiang (J. App. Polym. Sci. 2011, 122, 2309-2316), which can be interpreted as following. Part of the double peak appears because the platelet portion formed by PCL is partially melted, the unmelted portion can serve as a nucleation site to form a melt recrystallization, which can be melted at a higher temperature to form a molten double peak. Besides, thanks for the valuable comments on the DSC drawing in this article. We have revised it according to the comments.

Q4. The polymer structure in Fig 2 is not correct. Please revise it.

A4: Thanks very much for your professional suggestions. Upon your suggestions, we revised the incorrect polymer structure and replaced it.
